# Importance of vitamin D in acute and critically ill children with subgroup analyses of sepsis and respiratory tract infections: a systematic review and meta-analysis

Margarita Cariolou,[1] Meghan A Cupp,[1] Evangelos Evangelou,[1,2] Ioanna Tzoulaki,[1,2] Antonio J Berlanga-Taylor[1]

[1]MRC-PHE Centre for Environment and Health, Department of Epidemiology and Biostatistics, Imperial College London School of Public Health, London, UK
[2]Department of Hygiene and Epidemiology, University of Ioannina School of Medicine, Ioannina, Greece

**Correspondence to**
Dr Antonio J Berlanga-Taylor;
a.berlanga@imperial.ac.uk

## ABSTRACT

**Objectives** To estimate the prevalence of 25-hydroxyvitamin D (25(OH)D) deficiency and investigate its association with mortality in children with acute or critical conditions.

**Design** Systematic review and meta-analysis of observational studies.

**Data sources** PubMed, OVID, Google Scholar and the Cochrane Library searched until 21 December 2018.

**Eligibility criteria** Studies of children hospitalised with acute or critical conditions who had blood 25(OH)D levels measured.

**Data extraction and synthesis** We obtained pooled prevalence estimates of 25(OH)D deficiency and ORs for mortality. We calculated 95% CI and prediction intervals and investigated heterogeneity and evidence of small-study effects.

**Results** Fifty-two studies were included. Of 7434 children, 3473 (47.0%) were 25(OH)D deficient (<50 nmol/L). The pooled prevalence estimate of 25(OH)D deficiency was 54.6% (95% CI 48.5% to 60.6%, $I^2$=95.3%, p<0.0001). Prevalence was similar after excluding smaller studies (51.5%). In children with sepsis (18 studies, 889 total individuals) prevalence was 64.0% (95% CI 52.0% to 74.4%, $I^2$=89.3%, p<0.0001) and 48.7% (95% CI 38.2% to 59.3%; $I^2$=94.3%, p<0.0001) in those with respiratory tract infections (RTI) (25 studies, 2699 total individuals). Overall, meta-analysis of mortality (18 cohort studies, 2463 total individuals) showed increased risk of death in 25(OH)D deficient children (OR 1.81, 95% CI 1.24 to 2.64, p=0.002, $I^2$=25.7%, p=0.153). Four (22.0%) of the 18 studies statistically adjusted for confounders. There were insufficient studies to meta-analyse sepsis and RTI-related mortality.

**Conclusions** Our results suggest that 25(OH)D deficiency in acute and critically ill children is high and associated with increased mortality. Small-study effects, reverse causation and other biases may have confounded results. Larger, carefully designed studies in homogeneous populations with confounder adjustment are needed to clarify the association between 25(OH)D levels with mortality and other outcomes.

**Prospero registration number** CRD42016050638.

### Strengths and limitations of this study

► We comprehensively assessed the magnitude and relevance of vitamin D (25(OH)D) circulating levels in paediatric patients with acute and critical illness using a large number of studies with large total sample size with prespecified subgroup and sensitivity analyses.

► We used the currently recommended cut-off of less than 50 nmol/L for vitamin D deficiency.

► We did not find enough studies to perform meta-analyses for mortality from sepsis or respiratory tract infection in relation to vitamin D status.

► We did not identify longitudinal studies with multiple time point, preadmission or predisease vitamin D measurements.

► Most studies were single centre with heterogeneous patient groups and few controlled for important confounders that influence vitamin D levels such as age, body mass index, gender, season of measurements, vitamin D supplementation and comorbidities.

## INTRODUCTION

Vitamin D is an essential nutrient[1 2] representing a group of fat soluble secosteroids with key endocrine functions.[3] It is synthesised in the skin on sunlight exposure[4] while dietary sources, such as oily fish, egg yolk, certain fungi and supplements, are usually secondary sources. Vitamin D is critical in bone metabolism[5] and calcium homeostasis,[6] as well as acting as an important regulator in extraskeletal metabolic processes,[7] cardiovascular and immune systems.[8] Many observational and laboratory studies have observed the anti-inflammatory properties of vitamin D,[9] including direct regulation of endogenous antimicrobial peptide production.[10]

It is therefore crucial for humans to have sufficient vitamin D levels to maintain bone health and possibly improve response to

infection.[6 11 12] Infants and children are especially dependent on vitamin D to achieve healthy bone development and growth.[13 14] Well-known functional outcomes of adequate vitamin D levels in children include rickets prevention, higher bone mineral content and reduced bone fracture rates.[5 14] In otherwise healthy children in the USA, the reported prevalence of vitamin D deficiency (25(OH)D levels of <25 nmol/L) ranges from 9% to 18%.[15] The Endocrine Society Clinical Practice Guidelines and the Institute of Medicine (IOM) suggest that 25(OH)D levels less than 50 nmol/L (20 ng/mL) reflect a deficient state.[4 16]

Studies in adults reflect a high prevalence of vitamin D deficiency both in general intensive care unit (ICU) patients, and patients with sepsis, and strongly suggest an association between low vitamin D and poor clinical outcomes, including increased mortality, particularly in those suffering from sepsis.[2 17] Recent clinical trials of vitamin D supplementation in adults appear promising in both general critical care[18 19] and sepsis.[20]

Sepsis remains a challenging clinical entity with high social and economic costs.[21] Each year there are approximately 123 000 sepsis cases and around 37 000 deaths in England alone.[22] Recent reports show an increased prevalence of paediatric sepsis,[23] likely a reflection of an increased population with chronic comorbidities, higher rates of opportunistic infections and multidrug-resistant organisms.[24] Respiratory tract infections account for a large proportion of underlying diagnoses in acute and critical care conditions[24 25] but remain understudied.[26]

The magnitude, relevance and quality of evidence of vitamin D deficiency in children receiving acute care is not clear. Several recent studies have addressed these questions with mixed results. We sought to summarise the evidence regarding the implications of vitamin D deficiency and its prevalence in patients receiving general acute care, ICU patients, and patients with respiratory tract infection or sepsis in the paediatric population. We carried out a systematic review and meta-analysis of circulating vitamin D levels, as measured by 25(OH)D, to assess the prevalence of vitamin D deficiency (≤50 nmol/L) and its association with mortality in these conditions.

## METHODS
We used the Preferred Reporting Items for Systematic Reviews and Meta-Analyses (PRISMA) guidelines to report our review[27] (online supplementary table 1). We also followed the Meta-Analysis of Observational Studies in Epidemiology (MOOSE) guidelines[28] as no relevant randomised controlled trials have been reported.

### Search strategy and selection criteria
Our population of interest consists of paediatric patients with acute conditions and/or those treated in ICU or emergency units for acute conditions whose vitamin D status was assessed prior to or during admission. We included published cross-sectional, case–control and cohort studies that measured circulating 25(OH)D levels and either reported prevalence, ORs or data to enable calculation of these measures. Studies were excluded if they were reviews, case reports, surveys, commentaries, replies, not original contributions, experimental in vitro or if they recruited patients who were not treated in emergency, neonatal intensive care units (NICU), paediatric intensive care units (PICU) or for acute conditions. Studies were also excluded if they only enrolled vitamin D deficient patients, investigated healthy populations only or did not measure circulating 25(OH)D levels as an indicator of vitamin D status. When we identified more than one publication using the same cohort, we included the publication which shared our review's objective to investigate vitamin D levels and prevalence of deficiency.

For purposes of our review, we classified vitamin D deficiency as being 25(OH)D less than 50 nmol/L (equivalent to 20 ng/mL), as suggested by the IOM.[16] Different age categories were used to designate patients as 'children' in the studies reviewed. We therefore included all 'children' (neonates up to 21 years) as defined by each treating facility and this included 'neonates', 'infants', 'toddlers', 'children' and 'adolescents'.

We searched PubMed, OVID, Google Scholar and the Cochrane Library from inception up until 21 December 2018, with no language restrictions. Search terms used across these databases included: 'critical care', 'vitamin D', 'pediatric', 'child', 'neonate', 'toddler', 'intensive care unit', 'sepsis' and 'septic shock'. Search terms used in OVID and PubMed are listed in the online supplementary table 2A,B. Literature searches were performed by two investigators independently (MC and AJBT) and included initial screening of titles and abstracts, followed by full-text screening. Any disagreements for study eligibility were resolved by discussion between the two investigators. Reference lists of the selected papers, including reviews, were also checked for relevant titles. Abstracts of relevant titles were then assessed for eligibility. Corresponding authors were contacted to obtain additional information if necessary. A data extraction form was designed a priori in Excel. Variables extracted from each study included year of publication, country of study, clinical setting, cut-off given to define vitamin D deficiency, total number of children, total number of cases, study design and age range.

### Study quality assessment
The quality of each included study was assessed using the Newcastle-Ottawa Scale (NOS) adapted a priori for this review, for cohort, case–control and cross-sectional study designs (online supplementary table 3A–C).[29] We classified studies as low (1–3), medium (4–6) or high quality (7–9) for purposes of sensitivity analysis.

### Prevalence and mortality outcomes
In the majority of studies (n=40), prevalence of vitamin D deficiency was extracted as reported with a threshold of ≤50 nmol/L. If prevalence was not reported directly,

it was calculated using data provided in each study (cases ≤50 nmol/L per total number of study participants) (online supplementary table 4A,B). Extracted or calculated prevalence values were then combined in a meta-analysis. For mortality, we calculated unadjusted ORs as:

OR = (vitamin D deficient patients who died * vitamin D non-deficient patients who did not die)/ (vitamin D deficient patients who did not die * vitamin D non-deficient patients who died)

We had sufficient information to calculate ORs <50 nmol/L for 40 studies (77.0%). For the 12 studies with insufficient information, we used the lower cut-off values reported as a conservative approximation (online supplementary table 5). We converted 25(OH)D values using: nmol/L=ng/mL * 2.496.

## Data analysis

We obtained proportions of vitamin D deficiency with 95% CIs using the Clopper-Pearson method[30] in R. We used a random-effects model[31] to account for the variation observed within and between studies due to the different ages and acute conditions in the populations considered. For each meta-analysis we also obtained the 95% prediction interval (PI) to further account for between-study heterogeneity. This helps to evaluate how consistent an observed effect would be in a future study that will investigate the same association.[32] We obtained pooled proportions and pooled ORs with fixed-effects model for sensitivity analysis or in cases where heterogeneity was low.[33–35] For prevalence we also calculated median and IQR for comparisons with pooled prevalence estimates.

We investigated possible sources of heterogeneity using sensitivity and subgroup analyses. Cochran's Q was used to assess the heterogeneity and the $I^2$ statistic was used to estimate the percentage of total variation across studies which can be attributed to heterogeneity. CIs of $I^2$ were calculated to aid interpretation.[36] A Q value of <0.05 was considered significant and an $I^2$ statistic greater or equal to 75% indicated a high level of variation due to heterogeneity.[37 38] We used Egger's regression test to present results of small-study effects and funnel plot asymmetry[39] and generated funnel plots for visual assessment and screening. A p value <0.05 indicated evidence of small-study effects. With few studies, Egger's test has low power to detect such bias, therefore we only estimated small-study effects for analyses with more than 10 studies.[40] When small-study effects were detected based on this threshold, we used trim-and-fill methods to add potentially missed studies and recalculate an adjusted pooled estimate.[41]

To further assess heterogeneity, we used meta-regression to identify predictor variables that could explain variation in study prevalence estimates. We used restricted maximum likelihood estimations in the model to account for residual heterogeneity[42] and the Knapp-Hartung method to adjust CIs and test statistics. This method estimates between study variance using a t-distribution, rather than a z-distribution, yielding a more conservative inference.[43] We tested the following continuous predictors: year of study publication, total sample size and quality score. Categorical variables included study setting (PICU, NICU), study design (case–control, cross-sectional and cohort) and country group by geographic region and economic development (group 1, group 2 and group 3) and were dummy coded.

We used R V.3.5.0 and Microsoft Excel 2010 for analyses and data collection. The R packages 'meta'[44] and 'metafor'[45] were used for analyses. Only results of the random-effects model are reported for prevalence due to the expected heterogeneity between populations being considered. Our protocol is registered in PROSPERO (CRD42016050638).

## Role of the funding source

The study received funding from the UK Medical Research Council. The funders had no role in data collection, analysis, interpretation or writing of the report. All authors had access to the data in the study.

## Patient involvement

No patients were involved in this study. We only used data from previously published studies.

## RESULTS

### Screening and study characteristics

After title and abstract screening, we identified 2890 potentially relevant studies (figure 1) and 85 full-text articles were assessed for eligibility. Rationale for study exclusion included: studies including adults, study populations other than critically ill children or with acute conditions and studies of circulating vitamin D levels and deficiency in healthy children or in children with chronic conditions. Four studies[46–49] were excluded due to insufficient data reporting (online supplementary table 6). We also excluded three studies[50–52] that used the same cohort of children and included a single study to represent the cohort.[53] Ultimately, 52 studies met criteria for inclusion (online supplementary table 7).

The primary objective of most included studies was to determine circulating vitamin D concentration ('status') in children and/or prevalence of vitamin D deficiency. Secondary objectives included investigation of associations between deficiency of circulating vitamin D and various outcomes, such as hospital mortality length of stay, requirement of ventilation and/or illness severity (online supplementary table 8).

All included studies reported vitamin D measurement assay methods used (online supplementary table 9) and stated that samples were collected and analysed within the first 24 hours of hospital admission. Studies reported ethical approval and consent for participation from parents or guardians (online supplementary table 10). Included studies were published between 2004 and 2018,

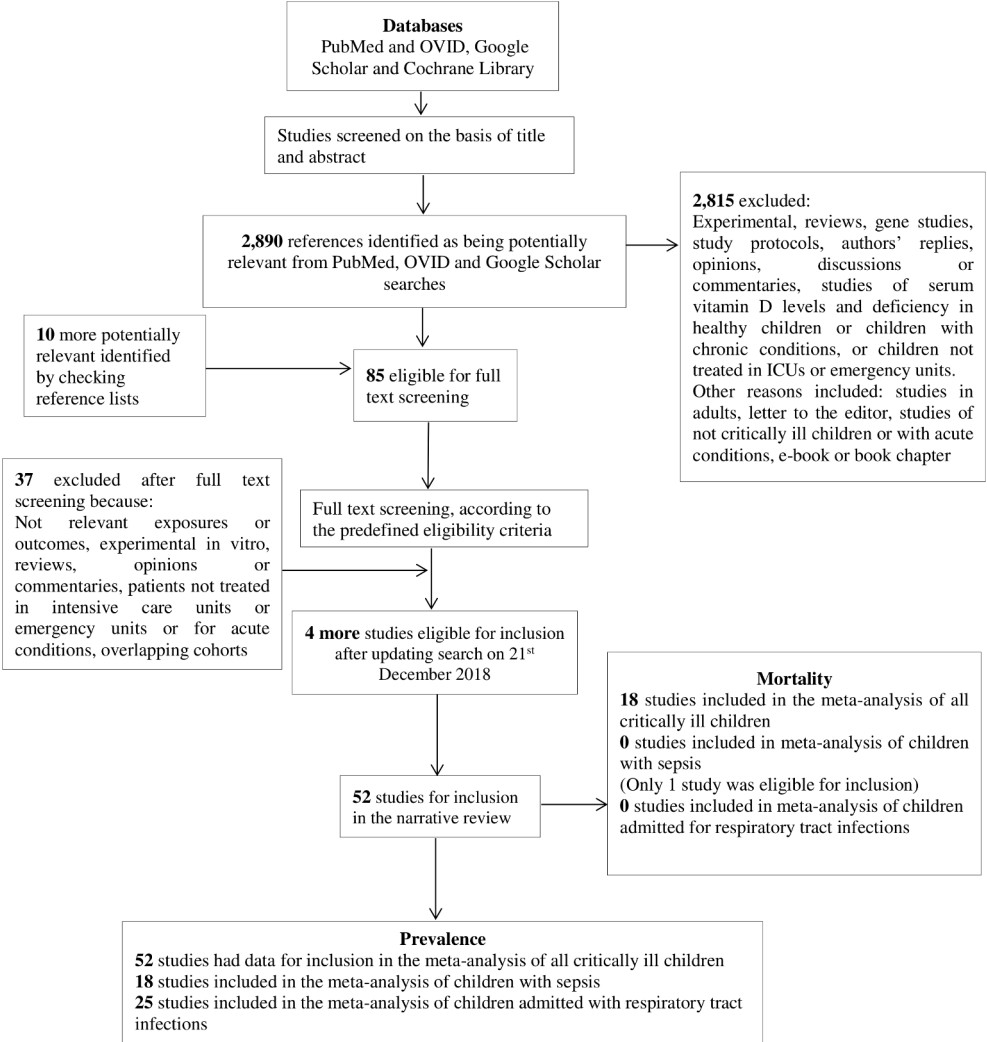

**Figure 1** Flow chart of study selection process. ICU, intensive care unit.

with the majority (n=40, 77.0%) published between 2014 and 2017 (online supplementary table 7). In total, 7434 children were hospitalised in PICU or NICU or emergency units or for acute conditions. Sample sizes of critically ill children ranged from 25[54] to 1016.[55] In 18 studies the total number of cases was greater than 100.

Studies originated from 15 countries, with the majority from India[8 56–65] (n=11) or Turkey[54 66–71] (n=7) (online supplementary table 7). All were of medium or high quality (NOS score median 7, range 4–8). The score range for cohort studies was 6–8 (n=30), for case–control studies 5–8 (n=18) and for cross sectional 4–6 (n=4). Studies used a broad range of ages to classify patients as 'children'. Seven studies (13.5%)[54 65 67 69–72] included only neonates. In two[67 72] of these studies, neonates were preterm. The largest age range was seen in the study of Ayulo *et al*[73], which included individuals between 1 and 21 years of age (online supplementary table 11). Forty-two of the included studies (80.8%) included patients admitted for medical conditions and the other 10[53 61 66 74–79] included both surgical and medical patients. Of the 52 included studies 26 used a control group and had a total

number of 2479 controls of which 773 (31.2%) were vitamin D deficient.

All studies included both female and male participants. For mortality, 4 of the 18 studies (22.0%) carried out multivariate regression analysis with adjustment for confounders. The remaining studies presented results using a variety of methods, including Spearman's correlation analysis, $X^2$ or Fisher's exact test or descriptive statistics.

### Prevalence of vitamin D deficiency

We included 52 studies representing a total of 7434 children hospitalised with critical or acute conditions. Of these, 3473 (47.0%) were classified as vitamin D deficient (<50 nmol/L). Prevalence of deficiency ranged from 5.0%[80] to 95.0%,[60] median (IQR) 56.3% (31.9%–75.2%) (online supplementary table 12). Sample sizes ranged from 25 to 1016, with a median of 82 individuals (online supplementary table 13). Using a random-effects model, the pooled prevalence estimate of vitamin D deficiency was 54.6% (95% CI 48.5% to 60.6%) with a high proportion of variation attributed to heterogeneity ($I^2$=95.3%, 95% CI 94.5% to 96.0%, p<0.0001) (figure 2) and

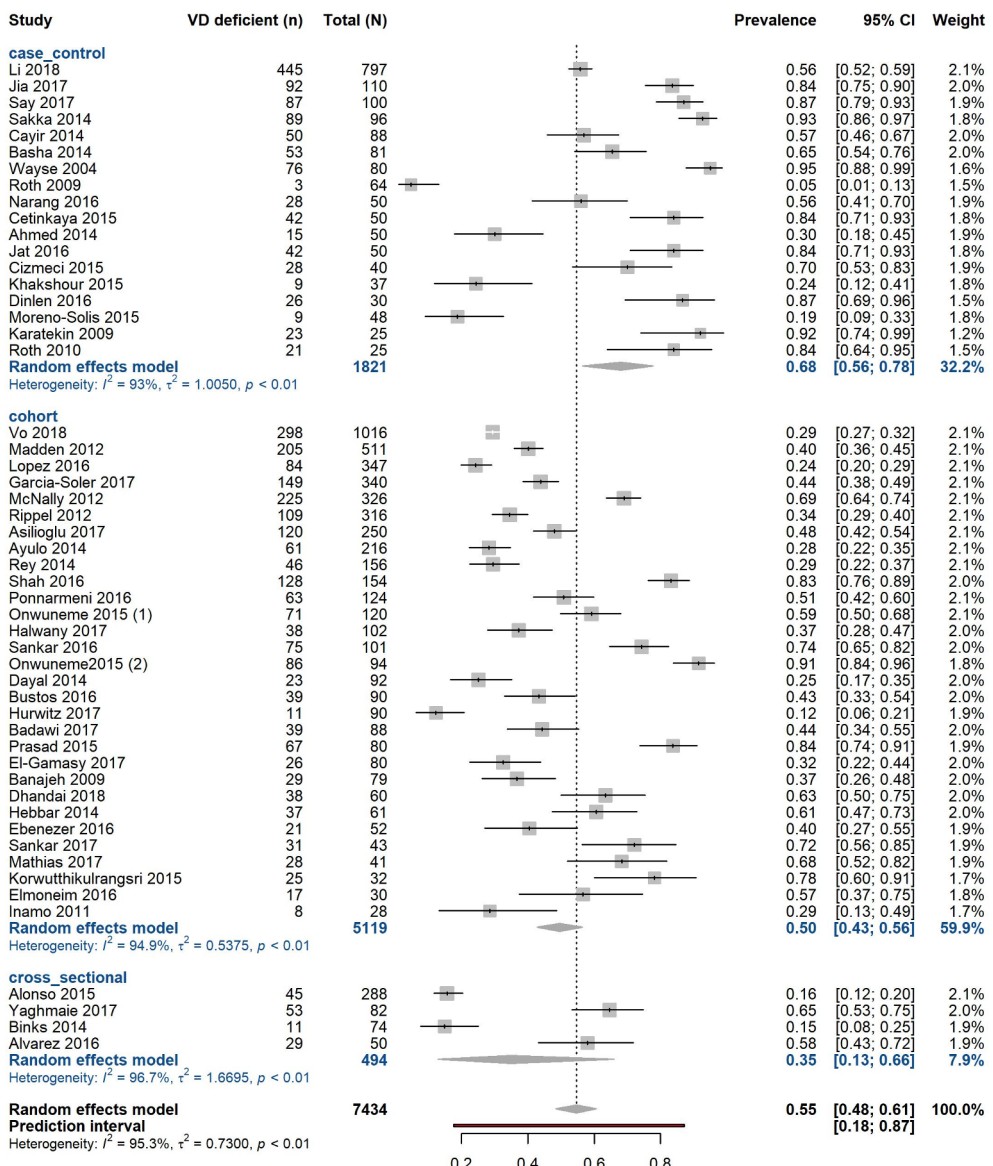

**Figure 2** Pooled prevalence estimate for vitamin D deficiency in acute and critically ill children by study design. Forest plot shows results from the random-effects model. Each diamond represents the pooled proportion of 25(OH)D deficiency for each of the subgroups (case–control, cohort, cross-sectional study designs). The diamond at the bottom represents the overall pooled proportion of all the 52 studies together. Each square shows the prevalence estimate of each study and the horizontal line across each square represents the 95% CI of the prevalence estimate. VD, vitamin D.

evidence of funnel plot asymmetry (p=0.01, Egger's test) (table 1 and online supplementary figure 1). Trim and fill analysis estimated 11 unpublished studies in the lower left-hand side of the funnel plot (online supplementary figure 1). The recalculated adjusted pooled estimate was lower 43.6% (95% CI 37.5% to 50.0%) with significant heterogeneity (p<0.0001).

### Sensitivity analysis for prevalence

We did not detect material differences in prevalence after exclusion of the 12 studies which did not directly report prevalence <50 nmol/L (53.0%, 95% CI 46.4% to 59.5%; $I^2$=95.5%, 95% CI 94.5% to 96.2%, p<0.0001) (online supplementary table 14).

When examining results by median sample size (defining 'large' as ≥82 and 'small' as <82), we found that

the 26[8 53 55–58 66–68 72–75 78 79 81–91] studies with larger sample size included 6094 total individuals and gave a prevalence estimate of 51.5% (95% CI 43.6% to 59.4%; $I^2$=96.8%, 95% CI 96.0% to 97.4%, p<0.0001). The remaining 26 studies with 'smaller' sample sizes included 1340 total children and estimated pooled prevalence as 58.2% (95% CI 47.5% to 68.2%; $I^2$=90.9%, 95% CI 87.9% to 93.2%, p<0.0001) (online supplementary table 14).

We also conducted analysis by study design. Cohort studies (n=30) yielded a prevalence estimate of 49.6% (95% CI 42.7% to 56.4%; $I^2$=94.9%, 95% CI 93.6% to 95.9%, p<0.0001). In case–control studies (n=18) the estimate was 68.1% (95% CI 56.5% to 77.8%; $I^2$=93.0%, 95% CI 90.4% to 94.9%, p<0.0001) and in cross-sectional (n=4) 34.8% (95% CI 12.8% to 66.0%; $I^2$=96.7%, 95% CI

**Table 1** Pooled estimates of vitamin D (25(OH)D) deficiency in acute and critically ill children and those with sepsis or respiratory tract infections

| Patient category | Number of studies (total number of individuals; number of deficient individuals) | Pooled proportion (%, 95% CI) Random effects | 95% PI | Pooled proportion (%, 95% CI) Fixed effects | Heterogeneity (I²) % (95% CI) | Q value df P value for heterogeneity | Egger's P value |
|---|---|---|---|---|---|---|---|
| All children (includes those with sepsis and respiratory tract infections) | 52 (7434; 3473) | 54.6 (48.5 to 60.6) | 17.5 to 87.2 | 45.7 (44.4 to 46.9) | 95.3 (94.5 to 96.0) | 1086.6 51 <0.0001 | 0.01 |
| Critically ill children with sepsis only | 18 (889; 565) | 64.0 (52.0 to 74.4) | 17.1 to 93.9 | 63.0 (59.3 to 66.6) | 89.3 (84.6 to 92.5) | 158.52 17 <0.0001 | 0.81 |
| Critically ill children with respiratory tract infections only | 25 (2699; 1076) | 48.7 (38.2 to 59.3) | 9.96 to 89.1 | 37.0 (35.0 to 39.1) | 94.3 (92.7 to 95.6) | 423.07 24 <0.0001 | 0.05 |

Vitamin D deficiency defined in our study as: 25(OH)D <50 nmol/L (20 ng/mL). $I^2$ statistic used to estimate heterogeneity between pooled studies: $I^2 \geq 75\%$ was considered high heterogeneity.

df, degrees of freedom; $I^2$, heterogeneity; PI, prediction interval.

94.0% to 98.2%, p<0.0001) (online supplementary table 14, figure 2).

We assessed whether studies' country of origin influenced results. Studies in India gave an estimate of 68.9% (95% CI 54.9% to 80.1%; $I^2$=96.7% (95% CI 94.0% to 98.2%, p<0.0001). Similarly, we found higher pooled prevalence estimates for studies from Turkey (76.3%, 95% CI 60.9% to 87.0%; $I^2$=91.1%, 95% CI 84.2% to 95.0%, p<0.0001). We also grouped studies by geography and economic development. Group 1: USA, Chile, Australia, Canada, Ireland, Japan, Spain; group 2: South Africa, China, Egypt, Iran, Turkey, Saudi Arabia; and group 3: Bangladesh, Thailand and India. Prevalence was 37.2% (95% CI 29.7% to 45.5%) for group 1 (n=20), 61.8% (95% CI 53.2% to 69.7%) for group 2 (n=19) and 70.8% (95% CI 58.3% to 80.7%) for group 3 (n=13) (online supplementary figure 2). Variation attributable to heterogeneity was still high in the three subgroups ($I^2$>90.0%).

Given the broad age range in included studies, we combined studies with only neonates[54 65 67 69–72] and observed a prevalence estimate of 83.0% (95% CI 73.1% to 89.8%) with less variation attributable to heterogeneity ($I^2$=76.6%, 95% CI 51.0% to 88.9%, p=0.0003). In all other studies (n=45) that included children of other age ranges, estimated prevalence was lower at 49.7% (95% CI 43.5% to 55.8%; $I^2$=95.2%, 95% CI 94.3% to 96.0%, p<0.0001) (online supplementary table 14 and figure 3).

### Post hoc investigation to determine sources of heterogeneity

To investigate the substantial heterogeneity observed in prevalence estimates, we incorporated study-specific characteristics (year of publication, total study sample size, quality score, study design, country group and clinical setting) as covariates in a random-effects meta-regression model. We identified clinical setting and country groups as significant predictors, p<0.01 (figure 3). We found that the model fitted with all available covariates can explain 32.9% of $I^2$ with $F$=4.57, p=0.001 (online supplementary table 15). We also conducted univariate meta-regressions for each of the six predictors (online supplementary figure 4).

### Prevalence of vitamin D deficiency in children with sepsis and in those with respiratory tract infections

A total of 889 (median 42, range 9–160) patients had a diagnosis of sepsis, of which 565 (63.5%) were vitamin D deficient. Sixteen of the 18 studies including patients with sepsis were cohort (88.9%) and two (11.1%) were case–control (online supplementary table 16). Most studies originated from India (n=7), Turkey (n=3) or Ireland (n=2) and 16 were published between 2014 and 2017. Thirteen studies took place in a PICU and the remaining[65 67 70 72] in NICUs. We found that all studies were of medium to high quality (median NOS score 7, range 6–8). Pooled prevalence of vitamin D deficiency was 64.0% (95% CI 52.0% to 74.4%) (figure 4) and median (IQR) 68.5% (50.4%–71.6%). Variation attributable to heterogeneity was high ($I^2$=89.3%, 95% CI 84.6%

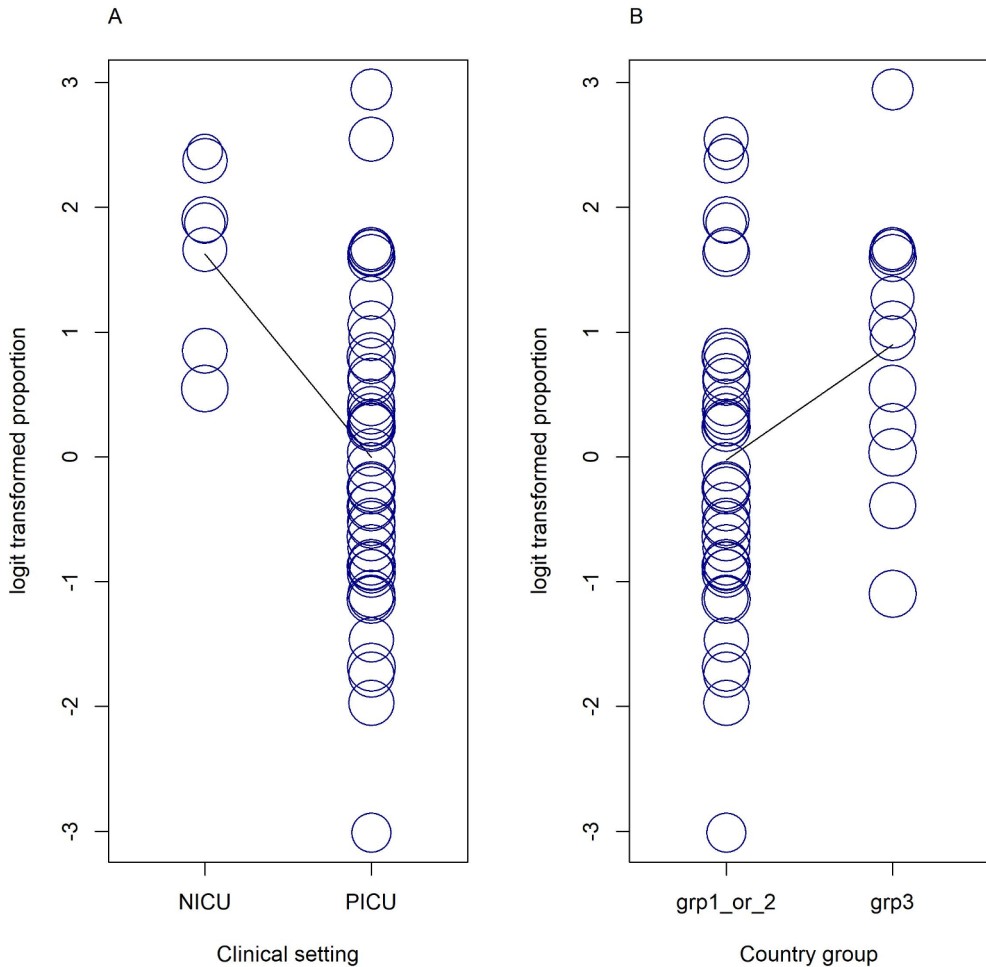

**Figure 3** Bubble plots of univariate meta-regressions. Each study is represented by a circle. Predictor variables: (A) clinical setting and (B) country groups are shown on the x-axis and the effect measure logit transformed proportion shown on the vertical (y-axis). Country group 1: USA, Chile, Australia, Canada, Ireland, Japan, Spain. Country group 2: South Africa, China, Egypt, Iran, Turkey, Saudi Arabia. Country group 3: Bangladesh, Thailand, India. grp, country group; NICU, neonatal intensive care unit; PICU, paediatric intensive care unit.

to 92.5%, p<0.0001). Funnel plot was symmetric (p>0.05) suggesting no small-study effects (p=0.81, Egger's test) (online supplementary figure 5).

We also separately analysed studies of patients admitted for respiratory tract infections (n=25) such as acute lower respiratory tract infection, pneumonia and bronchiolitis.

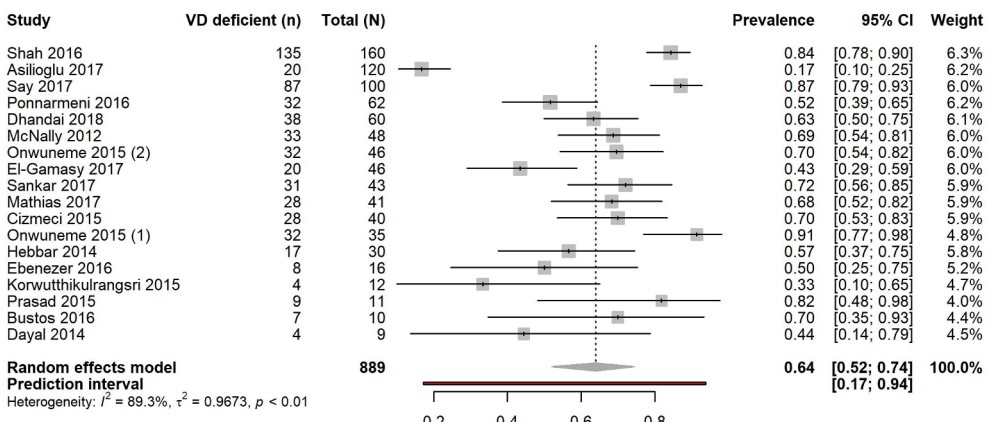

**Figure 4** Pooled prevalence estimate for vitamin D deficiency in children with sepsis. Forest plot shows result from the random-effects model. The diamond represents the overall pooled proportion of 25(OH)D deficiency from the meta-analysis of the 18 studies. Each square shows the prevalence estimate of each study and the horizontal line across each square represents the 95% CI of the prevalence estimate. VD, vitamin D.

Of these 2699 total individuals (median 50), 1076 (39.9%) were vitamin D deficient. These studies were of high to medium quality (median NOS score 7, range 6–8). Most originated from India (n=6) and Spain (n=4). We found a prevalence estimate of 48.7% (95% CI 38.2% to 59.3%; $I^2$=94.3%, 95% CI 92.7% to 95.6%, p<0.0001) and median (IQR) at 36.7% (24.3%–83.6%) with marginally non-significant evidence of bias (p=0.05, Egger's test) (table 1). We therefore applied the trim and fill method and obtained an adjusted pooled estimate of 37.4% (95% CI 27.6% to 48.4%) after four studies were added.

### Sensitivity analysis for prevalence in children with sepsis

Exclusion of the studies[64 67 72 92] using thresholds other than <50nmol/L for deficiency yielded a similar estimate of prevalence at 62.0% (95% CI 47.3% to 74.7%; $I^2$=89.7%, 95% CI 84.5% to 93.2%, p<0.0001) (online supplementary table 17).

We examined pooled prevalence estimates according to median sample size (<42 vs ≥42). Studies with a smaller sample size (n=9; 204 total individuals) showed a pooled prevalence estimate of 64.7% (95% CI 52.5% to 75.3%) with moderate variation attributable to heterogeneity ($I^2$=57.9%, 95% CI 11.8% to 79.9%, p=0.015). For the remaining nine studies (sample sizes ≥42, 685 total individuals) the estimate was 63.2% (95% CI 44.6% to 78.5%) with high variation attributable to heterogeneity ($I^2$=94.3%, 95% CI 91.1% to 96.3%, p<0.0001).

There was no material change in prevalence estimates when analysed according to study design. The 16 cohort studies (749 total individuals) gave an estimate of 61.4% (95% CI 48.6% to 72.8%) with high variation attributable to heterogeneity ($I^2$=88.8%, 95% CI 83.5% to 92.4%, p<0.0001). Case–control studies (n=2; 140 total individuals) showed a pooled prevalence of 80.0% (95% CI 58.8% to 91.8%; $I^2$=81.3%, 95% CI 20.5% to 95.6%, p<0.0001) (online supplementary table 17 and figure 6).

Studies from India (n=7) gave a prevalence estimate of 66.0% (95% CI 51.4% to 78.1%; $I^2$=81.1%, 95% CI 61.8% to 90.6%, p<0.0001). The three studies from Turkey assessing patients with sepsis gave a pooled estimate of 59.2% (95% CI 13.6% to 93.1%; $I^2$=97.8%, 95% CI 95.8% to 98.8%, p<0.0001) (online supplementary table 17).

The pooled prevalence estimate in the four studies[65 67 70 72] including neonates with sepsis was 73.7% (95% CI 60.3% to 83.8%; $I^2$=76.0%, 34.1–91.3, p=0.006). The 14 studies with children of different ages, excluding neonates, gave a pooled estimate of 60.7% (95% CI 45.5% to 74.0%); $I^2$=90.1%, 95% CI 85.2% to 93.4%, p<0.0001) (online supplementary table 17). Four of the studies[56 61 87 89] included children admitted with either sepsis or respiratory tract infections.

### Mortality in acute and critically ill children

We identified 18 cohort studies[8 53 56–59 61 64 66 72 73 75–79 89 92] assessing vitamin D status and mortality. These studies included a total of 2463 individuals, from which 220 deaths (17.2%) were observed in 1278 (51.9%) individuals with vitamin D deficiency and 99 deaths (8.4%) were observed in 1185 individuals without deficiency (48.1%).

All 18 studies took place in a PICU apart from one,[72] which considered only NICU patients. Sixteen of these studies (89.0%) were published between 2014 and 2017. Almost half (n=7) of the studies originated from India. Quality scores ranged from 5 to 8 with a median of 6.5.

Using a random-effects model, we found that vitamin D deficiency in critically ill children significantly increased the risk of death (OR 1.81, 95% CI 1.24 to 2.64, p=0.002) with low, non-significant heterogeneity ($I^2$=25.7%, 95% CI 0.0% to 58.0%, p=0.153) (figure 5). However, small-study effects cannot be easily excluded (p=0.084, Egger's test) (online supplementary figure 7) and the 95% PI (0.71–4.62) included the null value.

### Sensitivity analysis for mortality in acute and critically ill children

We obtained similar results through the fixed-effects model (OR 1.72, 95% CI 1.27 to 2.33, p=0.0005) (online supplementary figure 8). When excluding studies with thresholds other than <50nmol/L indicating deficiency, we found the association between vitamin D deficiency and increased risk of mortality still significant but lower,

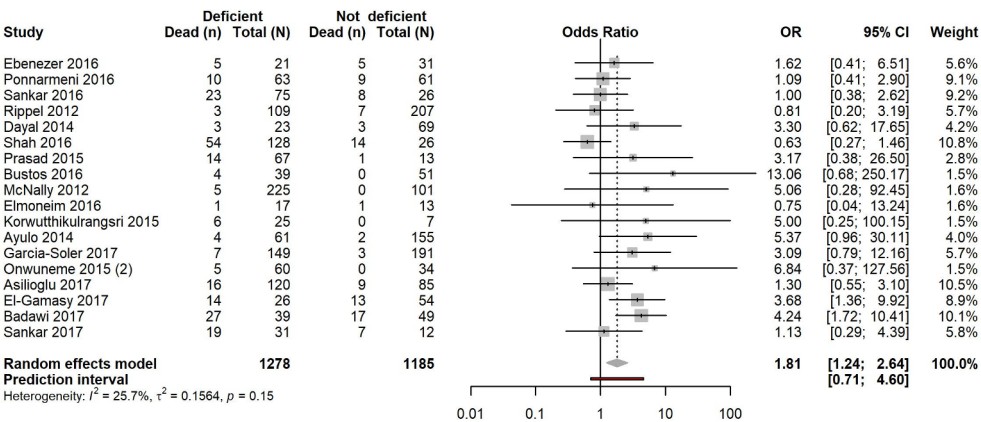

**Figure 5** Pooled OR of risk of mortality in vitamin D deficient versus vitamin D non-deficient acute and critically ill children. Forest plot shows result from the random-effects model. Diamond represents the overall OR (with corresponding 95% CI). Each square shows the OR of each study and the horizontal line across each square represents the 95% CI of the estimate.

both with the random (OR 1.59, 95% CI 1.05 to 2.41, p=0.028; I$^2$=24.3%, 95% CI 0.00% to 59.9%, p=0.191) and fixed-effects models (OR 1.52, 95% CI 1.08 to 2.13, p=0.016) without clear indication of small-study effects (p=0.120, Egger's test) (online supplementary table 18).

The association was positive but not significant when pooling the seven studies from India with the random-effects model (OR 1.08, 95% CI 0.70 to 1.69, p=0.710; I$^2$=0.0% 0.0–62.4, p=0.589) and similar with fixed effects (OR 1.08, 95% CI 0.70 to 1.69, p=0.710) (online supplementary table 18).

### Mortality in patients with sepsis and respiratory tract infections

We were unable to identify a sufficient number of studies assessing vitamin D and mortality for meta-analysis in individuals with sepsis. Three studies[8 64 67] measured vitamin D levels in paediatric patients with sepsis. One study[8] assessed mortality and did not find a significant association in children from 1 to 12 years with sepsis (n=124). None of the studies with children admitted for respiratory tract infections looked at the association between vitamin D deficiency and childhood mortality.

### DISCUSSION

Vitamin D deficiency is highly prevalent worldwide, even in countries with abundant sunshine. Studies have shown high prevalence of vitamin D deficiency in otherwise healthy children from high-income countries (9%–24%) but also from middle/low-income countries in the Indian subcontinent (36%–90%).[8]

We identified 52 studies representing a total of 7434 children treated in ICU or emergency units for acute conditions who had blood 25(OH)D levels measured close to or on admission. Our analysis shows that prevalence of vitamin D deficiency is generally high but very variable (range 5%[80]–95%[60]) across ICU and emergency units in the paediatric population, particularly in individuals with sepsis. Importantly, our analysis showed a significantly increased risk of mortality in critically ill children with vitamin D deficiency. We carried out several analyses for sensitivity including fixed-effects models, by study design, country group, age and sample size, and found generally consistent results. A recently published meta-analysis[93] also investigated prevalence of vitamin D deficiency in critically ill children and its association with risk of mortality and showed results similar to ours. The study did not clearly report heterogeneity and small-study effects, however, which we found to be critical limitations that must be addressed.

Subgroup analyses in patients with sepsis or respiratory tract infections demonstrated a high prevalence of vitamin D deficiency, consistent with the increased risk of bacterial or nosocomial infection in vitamin D deficient individuals identified elsewhere.[93]

Although sepsis is a leading cause of paediatric mortality and morbidity worldwide,[94] we found few studies assessing the relationship between vitamin D status and mortality in this population. We were unable to identify sufficient studies including patients with sepsis to perform a meta-analysis of vitamin D status and mortality. Sepsis remains an area of unmet need with high social and financial costs.[24] Diagnostic criteria,[95] a lack of adequate biomarkers[96] and targeted treatment remain important challenges in research on sepsis. We did not find studies that assessed the risk of mortality in relation to vitamin D deficiency in children admitted for respiratory tract infections either.

Strengths of our review include the large number of studies and large total sample size, allowing a high-powered investigation to identify meaningful associations. For our systematic review and meta-analysis, we followed prespecified eligibility criteria and used the PRISMA[27] and MOOSE guidelines[28] for reporting. We carried out multiple sensitivity analyses with few material differences in results. However, we note that the relationship between vitamin D deficiency and mortality was sensitive to study design and studies from India, probably due to the smaller number of individuals in those analyses. As expected for prevalence estimates, heterogeneity across studies was high overall. Only the prevalence analysis with neonates indicated somewhat lower variation attributable to heterogeneity (I$^2$=76.6%) along with a higher prevalence estimate (83.0%) compared with other analyses. We used meta-regression to investigate this substantial heterogeneity. From the six variables in our multivariable model, only clinical setting and country groups were found to be significant predictors of pooled prevalence estimates of vitamin D deficiency and the full model could explain 32.9% of heterogeneity (I$^2$). Studies in NICU yielded higher prevalence estimates compared with studies in PICU. Studies from group 3 countries were also associated with higher prevalence estimates compared with studies from countries of groups 1 and 2. Other variables, mainly individual patient characteristics such as age and ethnicity, were not directly available to us and may account for significant heterogeneity.

Our systematic review did not identify longitudinal studies with multiple time point, predisease or preadmission vitamin D measurements. The majority of studies were single centre with heterogeneous patient groups and relatively small sample sizes. Few studies accounted for important confounders that influence vitamin D levels such as age, gender, body mass index, season of measurements, vitamin D supplementation and comorbidities. The relationship observed between vitamin D deficiency and mortality could be due to reverse causation and future studies will need to control for covariates and other confounders. Low vitamin D levels could also represent a chronically deficient state due to reduced sunlight exposure, because of chronic illness, lifestyle factors or different country latitudes. In addition, we cannot rule out measurement bias such as dilution from intravenous fluids. Our results should be interpreted with caution since our review is based on evidence from observational studies. More research is warranted to strengthen the

evidence and investigate whether vitamin D could be causally linked to acute or critical illness and what its contribution might be through various mechanisms such as anti-inflammatory or antimicrobial peptide responses.

Although included studies were generally of good quality, sample sizes varied considerably and were typically small. Half of the studies included less than 100 cases and only 10 (19.2%) had a total sample size of more than 200 individuals. In addition, studies used a variety of definitions and age ranges to designate individuals as children. Our analysis only included mortality as a clinical outcome. A further general limitation is the difference in thresholds for vitamin D deficiency, particularly in the levels which are considered normal for infants and young children. Our assessment used the currently recommended threshold for deficiency (25(OH)D ≤50 nmol/L)[16] and a conservative estimate for studies which used different criteria. Although our review included a large number of studies and individuals, all studies were observational, and results could be subject to small-study effects.

Vitamin D remains an attractive biomarker and potential therapeutic agent in patients with acute and critical care conditions. Our review suggests that high-quality focused studies in each relevant paediatric population are needed first, which could then be followed by trials to establish safety and appropriate treatment regimens in children with acute or critical illness.

**Contributors** AJBT conceived the study. AJBT and IT designed the study. MC collected data and performed the analysis with input from MAC, IT, AJBT and EE. MC and AJBT wrote the manuscript with contributions from all authors.

**Funding** AJBT was supported by the Medical Research Council (UK MED-BIO Programme Fellowship, MR/L01632X/1).

**Competing interests** None declared.

**Patient consent for publication** Not required.

**Provenance and peer review** Not commissioned; externally peer reviewed.

**Data sharing statement** Data and computational code used for processing and analysis are available at https://github.com/margarc/VitaminD_children

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
