## [Reviewer comments · BMJ Open]

ARTICLE DETAILS

TITLE (PROVISIONAL)	Importance of vitamin D in acute and critically ill children with subgroup analyses of sepsis and respiratory tract infections: a systematic review and meta-analysis
AUTHORS	Cariolou, Margarita; Cupp, Meghan; Evangelou, Evangelos; Tzoulaki, Ioanna; Berlanga-Taylor, Antonio

VERSION 1 - REVIEW

REVIEWER	Reed Siemieniuk McMaster University, Canada
REVIEW RETURNED	18-Nov-2018

GENERAL COMMENTS	The results of the meta-analysis of prevalence studies results in a pooled estimate that is difficult to understand. There is little justification to weigh larger studies more than smaller ones when there are almost certainly idiosyncratic differences between locations and populations that explain the differences between studies. The pooled point estimate is vulnerable to over-influence from idiosyncratic populations. The median and a measure of distribution (e.g., IQR) is probably more reliable way to measure the 'typical' prevalence. Fixed effects meta-analysis does not "avoid false conclusions that could result from small-study effects." All of the available methods that aim to overcome small study effects have serious limitations, but the trim and fill method is one reasonable option. Excluding small studies as the authors have done in a sensitivity analysis is also a reasonable option that probably provides a more credible point estimates (and may be worth presenting in the abstract rather than the less credible estimate from studies of all sizes). As expected, there seems to large differences in prevalence of vitamin D deficiency by geographic region. The overall estimate is therefore meaningless in that it is a reflection of the relative number of studies performed in higher vs. lower prevalence areas. I suggest highlighting the different prevalences in the specific settings and locations rather than the overall estimate. There is evidence of small study effects (probably due to publication bias), suggesting that the pooled point estimate is not a reliable measure of overall prevalence of vitamin D deficiency.
--

	For the purposes of defining vitamin D deficiency and its effect on hospital outcomes, the authors seem to be confusing study design types. Many of the studies classified as case-control studies appear to be cohort studies (e.gs., Rey 2014; Ponnarmeni, 2016). Why do the authors categorize small vs. large studies as <100 vs. >100 in one section, and <40 vs. >40 in another? Ideally, the cutoff would have been defined prior to doing the analyses and presenting the data. In the abstract, presenting the confidence intervals around what I presume is the I2 is confusing without more explanation. The authors should also clarify whether the p-value is for treatment effect or for heterogeneity (what is the null hypothesis?).
--	---

REVIEWER	José Moreira Instituto Nacional de Infectologia Evandro Chagas (INI/FIOCRUZ), Rio de Janeiro, Brasil
REVIEW RETURNED	23-Nov-2018

GENERAL COMMENTS	Dear Editor Thank you for inviting me to review the manuscript entitle “Importance of vitamin D in critically ill children with subgroup analyses of sepsis and respiratory tract infections: a systematic review and meta-analysis” submitted to your esteemed Journal. Cariolou and colleagues aim to estimate the pooled prevalence of vitamin D deficiency and its association with mortality in critically ill children admitted at PICU, with subgroup analysis for those with sepsis and those with respiratory infections. I was asked to give constructive feedback with an emphasis on the statistical methods and analyses used; however, I felt interested in adding some non-statistical comments that would increase the manuscript quality. Overall the manuscript is well-written and scientifically sound. Recent evidence has shown that critically ill children had a high prevalence of vitamin D deficiency (i.e., defined as 25(OH) D below 50 nmol/L), and this deficiency is strongly associated with all-cause mortality. Interestingly, the few intervention trials conducted so far, have shown that vitamin D supplementation reduced mortality in general critically ill population. Non-statistical comments Definitions  • Considering the heterogenic definition used in the studies to define children, could you specify which range of age do you accept for the review? • In line 52, page 5, the authors state that they included studies enrolling pediatric patients with acute conditions and/or treated in ICU or emergency units for acute conditions. Could you elaborate on the definition of acute conditions? Studies in which researchers reported on populations admitted in emergency departments were included? If so, we might assume that a portion of the included population is not in a critical state and is not the primary target of the review.
--

	Data extraction • In line 3, page 7, the authors state that a data extraction form was designed a priori. However it essential to describe in sufficient details some aspects related to data extraction. How was data extracted from each study (electronically or manually)? What were the main variables of interest? Did you pilot the CRF before initiating the actual review process? Outcomes • I understand that mortality is a hard outcome and must be investigated. However, other non-negligible outcomes could be explored as well because they might be relevant from a clinical perspective. I would suggest those following outcomes (if data are available in the majority of included studies): mechanic ventilation use, vasopressor support and ICU length of stay. Results • Concerning the study characteristics, could the authors comment on the type of patients included in the observational studies (admitted for surgical/medical conditions) and the number of studies that also evaluated the vitamin D status in a control group? In case do exist control groups, what was the proportion of vitamin D deficiency in the control groups?• On page 14, the authors describe a sub-analysis regarding the prevalence of vitamin D deficiency in children with sepsis and those with respiratory infections. However, there is a massive overlapping between those conditions and I would suggest that authors conduct a sub-analysis ONLY in those with sepsis at admission (and discuss the limitations of the potentially different sepsis definitions used in each study). Sepsis is a good disease surrogate of severe infection with organ dysfunction both in developing and developed settings. Remember that the main focus of infection in septic patients is the respiratory tract, so the effect of the latter group in vitamin D status could still be captured. Statistical comments • In addition to I2, report indices that do report the dispersion of true effects on an absolute scale; these are the indices that address the questions that clinicians are much interested in (i.e., how much the vitamin D deficiency prevalence varies within the population?). Please estimate the prediction interval for prevalence (when the effect size is prevalence) and prediction interval for odds ratios (when the effect size is odds ratio). I2 is not an absolute measure of heterogeneity in a meta-analysis. It tells us nothing about the range of effect, and it does not tell us how much the effect size varies; instead it tells us what proportion of the observed variance would remain if we could eliminate the sampling error. If we are interested in determining how much the true effects sizes vary, we are looking for an amount, and not a proportion (which is the case for I2)• Regarding the meta-regression model for vitamin D deficiency, please add a table with all the included moderator variables and their respective beta coefficients, confidence intervals, p-values and the standard errors resulting from the meta-regression.
--	--

VERSION 1 – AUTHOR RESPONSE

Reviewer(s)' Comments to Author:

Reviewer: 1

Reviewer Name: Reed Siemieniuk

Institution and Country: McMaster University, Canada

Please state any competing interests or state 'None declared': None declared

Please leave your comments for the authors below

- The results of the meta-analysis of prevalence studies results in a pooled estimate that is difficult to understand. There is little justification to weigh larger studies more than smaller ones when there are almost certainly idiosyncratic differences between locations and populations that explain the differences between studies. The pooled point estimate is vulnerable to over-influence from idiosyncratic populations. The median and a measure of distribution (e.g., IQR) is probably more reliable way to measure the 'typical' prevalence.

Response: Many thanks for the detailed questions and suggestions. Although much research has been carried out, it remains largely unclear the variety of effects and magnitude that vitamin D may have, particularly in paediatric groups in intensive and acute care. We agree that there is high heterogeneity in the included studies, particularly among populations and age groups. We also agree that pooled estimates may be influenced by particular groups. We carried out analyses using weighting with random effects models and several sensitivity analyses to this effect. We also performed meta-regression to investigate possible sources.

As suggested by the reviewer, we have now calculated the median and interquartile range in estimates of the main analyses. We agree these provide further information and have quoted the median (IQR) in tables of results for comparison purposes and to give additional insights from the data.

- Fixed effects meta-analysis does not “avoid false conclusions that could result from small-study effects.” All of the available methods that aim to overcome small study effects have serious limitations, but the trim and fill method is one reasonable option. Excluding small studies as the authors have done in a sensitivity analysis is also a reasonable option that probably provides a more credible point estimates (and may be worth presenting in the abstract rather than the less credible estimate from studies of all sizes).

Response: We have revised this sentence to “We obtained pooled proportions and pooled ORs with fixed effect model for sensitivity analysis or in cases where heterogeneity was low” on page 8 lines 11-13. Although the trim and fill method also has limitations we followed the suggestion by the reviewer and used it when the Egger’s test was significant (according to our cut off $p < 0.05$) to get an estimate that adjusts for missing studies. We added the use of this method, on page 9 lines 2-3.

See updated “Additional Figure 1” and also see page 12 lines 6-9, page 15 lines 17-19 and page 18 lines 6-7 for results after trim and fill method was applied. We note that other authors use $p < 0.10$. If we had used $p < 0.10$ as cut point then more results would be estimated to show bias.

We now added the result of the sensitivity analysis that excludes smaller studies in the abstract. The pooled prevalence estimate excluding small studies was very similar (51.5%) to the pooled prevalence estimate with all 52 studies (54.6%).

- As expected, there seems to be large differences in prevalence of vitamin D deficiency by geographic region. The overall estimate is therefore meaningless in that it is a reflection of the relative number of studies performed in higher vs. lower prevalence areas. I suggest highlighting the different prevalences in the specific settings and locations rather than the overall estimate.

Response: Although we generally agree we felt it was important to first perform a global analysis given the many reports of worldwide deficiency. Following this however, including stating its caveats and heterogeneity, we report the results of analyses for country groups and clinical settings on page 13 lines 15-21, page 14 lines 1-7. We obtained the overall pooled estimate and then separately in each country group to make comparisons.

- There is evidence of small study effects (probably due to publication bias), suggesting that the pooled point estimate is not a reliable measure of overall prevalence of vitamin D deficiency.

Response: It is common to identify evidence of small study effects due to publication or other biases in meta-analyses results unfortunately (Sterne 2000). We investigated whether there was evidence of small study effects both statistically using the Egger's regression asymmetry test (using $p < 0.05$ as threshold for significance) and graphically with funnel plots. As suggested, we also used the trim and fill method to adjust (examine) possible biases. We have now clarified that we used $p < 0.05$ as a threshold for significance on page 8 lines 20-22 but have cautioned against biased interpretation as small-study effects cannot be excluded and some of our p-values are near or under 0.10 and 0.05. We agree that validity can be questioned because of both heterogeneity and small-study effects and have included this in the Abstract and Discussion.

- For the purposes of defining vitamin D deficiency and its effect on hospital outcomes, the authors seem to be confusing study design types. Many of the studies classified as case-control studies appear to be cohort studies (e.g.s., Rey 2014; Ponnarmeni, 2016).

Response: Authors of some studies (such as Rey 2014 and Ponnarmeni 2016) classified their studies as being "prospective case-control" or "prospective observational" with case-control groups. Despite their classifications we judged that "case-control" is the best way to describe the design and methodology of these studies and therefore classified them as such.

- Why do the authors categorize small vs. large studies as <100 vs. >100 in one section, and <40 vs. >40 in another? Ideally, cutoff would have been defined prior to doing the analyses and presenting the data.

Response: The difference in categorization of "small" versus "large" studies was to reflect the differences in total sample size in each analysis (i.e. overall and the analysis with septic children only). We agree with the reviewer that those cut-offs were arbitrary, and we have now calculated the median number of children to categorise studies as "small" versus "large" and repeated the analyses. Results have been updated accordingly in text on page 13 lines 4-9 and page 16 lines 1-6. Also see Additional Tables 14 and 17 where results have been updated. Results are very similar however as previous values were close to the median in both cases.

- In the abstract, presenting the confidence intervals around what I presume is the I² is confusing without more explanation. The authors should also clarify whether the p-value is for treatment effect or for heterogeneity (what is the null hypothesis?).

Response: The p-value presented is for heterogeneity. The null hypothesis was that there is not significant heterogeneity in summary estimates. We used a p-value of less than 0.05 to indicate that there is significant heterogeneity (alternative hypothesis). We deleted the 95% confidence interval of heterogeneity from the abstract to avoid confusion, but these are still quoted in text. We have also specified that the p-value is for heterogeneity in all Tables where heterogeneity results are reported (see updated Table 1, Additional Table 14 and Additional Table 17).

Reviewer: 2

Reviewer Name: José Moreira

Institution and Country: Instituto Nacional de Infectologia Evandro Chagas (INI/FIOCRUZ), Rio de Janeiro, Brasil

Please state any competing interests or state 'None declared': None declared

Please leave your comments for the authors below

Please find below my comments for the authors (see attachment).

Dear Editor

Thank you for inviting me to review the manuscript entitle "Importance of vitamin D in critically ill children with subgroup analyses of sepsis and respiratory tract infections: a systematic review and meta-analysis" submitted to your esteemed Journal. Cariolou and colleagues aim to estimate the pooled prevalence of vitamin D deficiency and its association with mortality in critically ill children admitted at PICU, with subgroup analysis for those with sepsis and those with respiratory infections.

I was asked to give constructive feedback with an emphasis on the statistical methods and analyses used; however, I felt interested in adding some non-statistical comments that would increase the manuscript quality.

Overall the manuscript is well-written and scientifically sound. Recent evidence has shown that critically ill children had a high prevalence of vitamin D deficiency (i.e., defined as 25(OH) D below 50 nmol/L), and this deficiency is strongly associated with all-cause mortality. Interestingly, the few intervention trials conducted so far, have shown that vitamin D supplementation reduced mortality in general critically ill population.

Response: Many thanks for your constructive comments and suggestions for the improvement of our manuscript.

Non-statistical comments

Definitions

- Considering the heterogenic definition used in the studies to define children, could you specify which range of age do you accept for the review?

Response: Our selection criteria included all ages defined as paediatric, this includes neonates to children less than 21 years. We now clarified this age range on page 6 lines 15-17.

- In line 52, page 5, the authors state that they included studies enrolling paediatric patients with acute conditions and/or treated in ICU or emergency units for acute conditions. Could you elaborate on the definition of acute conditions? Studies in which researchers reported on populations admitted in emergency departments were included? If so, we might assume that a portion of the included population is not in a critical state and is not the primary target of the review.

Response: Our study criteria included both acute and critically ill individuals. We have clarified the wording in the title, abstract and throughout the manuscript to reflect that our target population were children in acute respiratory or critical conditions.

Data extraction

- In line 3, page 7, the authors state that a data extraction form was designed a priori. However it essential to describe in sufficient details some aspects related to data extraction. How was data extracted from each study (electronically or manually)? What were the main variables of interest? Did you pilot the CRF before initiating the actual review process?

Response: We searched for relevant information both using key words electronically and manually and then data was extracted in Excel spreadsheets. Variables of interest from each study included year of publication, country of study, clinical setting, cut-off given to define vitamin D deficiency, total number of children, total number of cases, study design and age range. This information is now added on page 7 lines 4-7. Our review protocol is registered in PROSPERO (see page 3 line 9).

Outcomes

- I understand that mortality is a hard outcome and must be investigated. However, other non-negligible outcomes could be explored as well because they might be relevant from a clinical perspective. I would suggest those following outcomes (if data are available in the majority of included studies): mechanic ventilation use, vasopressor support and ICU length of stay.

Response: Although we agree that further clinical outcomes would be desirable to study, we chose to focus on prevalence and mortality with sub-group analysis as pre-specified primary outcomes. Other groups have recently published similar findings which overlap with ours and include other outcomes of interest (See ref. 93 McNally 2017 Crit Care). We discuss this on page 19 lines 12-18.

Results

- Concerning the study characteristics, could the authors comment on the type of patients included in the observational studies (admitted for surgical/medical conditions) and the number of studies that also evaluated the vitamin D status in a control group? In case do exist control groups, what was the proportion of vitamin D deficiency in the control groups?

Response: These are important questions which we detail in supplementary tables "Additional Table 7" and "Additional Table 13". Briefly, 42 of the included studies (80.8%) included patients admitted for medical conditions and the other 10 had both surgical and medical patients. Of the 52 included studies, 26 used a control group and had a total number of 2,479 controls of which 773 (31.2%) were vitamin D deficient. We added this information on page 11 lines 11-14.

- On page 14, the authors describe a sub-analysis regarding the prevalence of vitamin D deficiency in children with sepsis and those with respiratory infections. However, there is a massive overlapping between those conditions and I would suggest that authors conduct a sub-analysis ONLY in those with sepsis at admission (and discuss the limitations of the potentially different sepsis definitions used in each study). Sepsis is a good disease surrogate of severe infection with organ dysfunction both in developing and developed settings. Remember that the main focus of infection in septic patients is the respiratory tract, so the effect of the latter group in vitamin D status could still be captured.

Response: We performed these sub-group analyses independently. See page 15 lines 3-12 for main results of sepsis analysis and lines 15-23 for those with respiratory tract infections. We now clarified this on page 15 line 11 and edited column 1 of Table 1 to clearly reflect this. Included articles had clear diagnostic criteria for either sepsis or respiratory tract infections (excluding sepsis). The only papers that had children both with sepsis and with acute respiratory tract infections were Shah 2016, Bustos 2016, Ebenezer 2016 and Li 2018 but clearly distinguished between these two populations.

In all the included papers, sepsis is diagnosed at admission or in the first hours after admission. We therefore believe that this association is covered in our sub-analysis of sepsis. The definitions given for sepsis in each study are listed in Additional Table 4B. As we discuss on page 19 line 22 and page 20 lines 1-2 sepsis is a complex clinical syndrome and the different sepsis definitions add to heterogeneity.

Statistical comments

- In addition to I², report indices that do report the dispersion of true effects on an absolute scale; these are the indices that address the questions that clinicians are much interested in (i.e., how much the vitamin D deficiency prevalence varies within the population?). Please estimate the prediction interval for prevalence (when the effect size is prevalence) and prediction interval for odds ratios (when the effect size is odds ratio). I² is not an absolute measure of heterogeneity in a meta-analysis. It tells us nothing about the range of effect, and it does not tell us how much the effect size varies; instead it tells us what proportion of the observed variance would remain if we could eliminate the sampling error. If we are interested in determining how much the true effects sizes vary, we are looking for an amount, and not a proportion (which is the case for I²)

Response: We have now added prediction intervals both for estimates of prevalence and mortality (effect size: odds ratio). We have also added a separate column in results tables to indicate 95% prediction interval (PI) for each summary estimate. Prediction intervals were also added to forest plots of random effects models (see updated Figures, both Main and Additional). We have also updated the Methods, Data Analysis section page 8 lines 9-11 and Abstract to reflect this.

- Regarding the meta-regression model for vitamin D deficiency, please add a table with all the included moderator variables and their respective beta coefficients, confidence intervals, p-values and the standard errors resulting from the meta-regression.

Response: These had already been included in our first submission (i.e. moderator variables, respective beta coefficients, confidence intervals, p-values and the standard errors). Please see Additional Table 15.

VERSION 2 – REVIEW

REVIEWER	Reed Siemieniuk McMaster University, Canada
REVIEW RETURNED	15-Feb-2019

GENERAL COMMENTS	Thank you for carefully considering my suggestions previously. I think that the manuscript is almost ready for acceptance. The only thing that I believe needs to be corrected is the study designs. I continue to disagree with the authors that the studies they currently call 'case-control' are in fact case control studies. Case-control studies cannot be used to calculate prevalence, because the 'cases' do not have a denominator. The risk of bias tool specific to case-control studies is inappropriate for prevalence studies. The authors should use a more appropriate risk of bias tool.
---

REVIEWER	Jose Moreira Instituto Nacional de Infectologia Evandro Chagas, Fundacao Oswaldo Cruz (FIOCRUZ), Brazil
REVIEW RETURNED	08-Feb-2019

GENERAL COMMENTS	The authors have addressed all my concerns in this revised version. I recommend the publication in the current format.
--

VERSION 2 – AUTHOR RESPONSE

Reviewer(s)' Comments to Author:

Reviewer: 2

Reviewer Name: Jose Moreira

Institution and Country: Instituto Nacional de Infectologia Evandro Chagas, Fundacao Oswaldo Cruz (FIOCRUZ), Brazil

Please state any competing interests or state 'None declared': None declared.

Please leave your comments for the authors below

The authors have addressed all my concerns in this revised version. I recommend the publication in the current format.

Response: Many thanks for your recommendation and helpful suggestions.

Reviewer: 1

Reviewer Name: Reed Siemieniuk

Institution and Country: McMaster University, Canada

Please state any competing interests or state 'None declared': None declared

Please leave your comments for the authors below

Thank you for carefully considering my suggestions previously. I think that the manuscript is almost ready for acceptance.

Response: Many thanks, we have updated our analysis and further improved the manuscript. Please see below.

- The only thing that I believe needs to be corrected is the study designs. I continue to disagree with the authors that the studies they currently call 'case-control' are in fact case control studies. Case-control studies cannot be used to calculate prevalence, because the 'cases' do not have a denominator.

Response: We agree that the epidemiological study designs can be difficult to define in certain cases. We have reviewed these studies again and concentrated on the aims, methodology and outcomes of each study. We re-classified eight studies (including Rey 2014 and Ponnarmeni 2016 which were

indicated by the reviewer in the first revision) from case-control to cohort. All the necessary analyses have therefore been repeated and results have been updated accordingly without any significant differences compared to our previous findings. As necessary, we have also updated Figure 2, Figure 3, Additional Figure 4 and Additional Figure 6. As we have indicated in our methodology page 6 lines 3-5 and page 7 lines 14-16, we have included observational studies which either reported the prevalence of vitamin D deficiency directly (i.e. proportion/percentage of vitamin D deficient acute or critically ill children over the total number of acute or critically ill children) or provided data that enabled calculation of this proportion.

- The risk of bias tool specific to case-control studies is inappropriate for prevalence studies. The authors should use a more appropriate risk of bias tool.

Response: We acknowledge that risk of bias or methodological quality assessment remains challenging since all existing tools are not without limitations and there is currently no agreed 'gold standard' appraisal tool. We have carefully considered the reviewer's comment and attempted to identify if there is a more appropriate tool than the one we used in our study. We found that tools used by other authors have similarities with the Newcastle-Ottawa Scale. For example, the methodological quality critical appraisal checklist proposed by the Joanna Briggs Institute Group¹ also assesses sample representativeness, recruitment appropriateness, comparability of cases and controls with respect to data analysis, confounding factors, exposure ascertainment and measurement. The quality assessment checklist for prevalence studies adapted from Hoy et al² also considers similar items including sample representativeness and selection, case definition and response rate, amongst others. Overall, there is a large overlap between tools without a clear distinction between them.

The Newcastle-Ottawa Scale assesses the quality of non-randomised studies in meta-analyses, including case-control and cohort studies. It has been used in multiple studies and has been refined based on experience of users. It can be adapted to accommodate studies of prevalence.^{3, 4} Multiple reviews of observational studies that investigated prevalence used Newcastle-Ottawa Scale, see some examples in the reference list below.⁵⁻¹²

We modified items of the Newcastle-Ottawa Scale according to the topic of our review a priori, and we have now clarified this in text page 7 lines 9-11. There is clearly considerable interest but despite the proliferation of tools there is still somewhat limited evidence regarding validation and comparability. We would welcome future work that evaluated the comparativeness and utility of the many quality and risk of bias tools available to avoid duplication and ensure appropriate standards are used.

References

1. Munn Z, Moola S, Riitano D, et al. The development of a critical appraisal tool for use in systematic reviews addressing questions of prevalence. *Int J Health Policy Manag* 2014;3(3):123-8.
2. Hoy D, Brooks P, Woolf A, et al. Assessing risk of bias in prevalence studies: modification of an existing tool and evidence of interrater agreement. *J Clin Epidemiol* 2012;65(9):934-9.
3. Wells GA, SB OCD, Peterson J, Welch V, Losos M, Tugwell P. The Newcastle-Ottawa Scale (NOS) for assessing the quality of nonrandomised studies in meta-analyses [Available from: http://www.ohri.ca/programs/clinical_epidemiology/oxford.asp.
4. Deeks JJ, Dinnes J, D'Amico R, et al. Evaluating non-randomised intervention studies. *Health Technol Assess* 2003;7(27).
5. Pampena R, Kyrgidis A, Lallas A, et al. A meta-analysis of nevus-associated melanoma: Prevalence and practical implications. *J Am Acad Dermatol* 2017;77(5):938-45. e4.

6. Dachew BA, Bifftu BB, Tiruneh BT, et al. Prevalence and determinants of mental distress among university students in Ethiopia: a systematic review protocol. *Systematic Reviews* 2019;8(1):47.
7. Qiu L, Xia W, Li W, et al. The prevalence of microsporidia in China : A systematic review and meta-analysis. *Sci Rep* 2019;9(1):3174.
8. Tankeu AT, Bigna JJ, Nansseu JR, et al. Prevalence and patterns of congenital heart diseases in Africa: a systematic review and meta-analysis protocol. *BMJ Open* 2017;7(2):e015633.
9. Alinaghi F, Calov M, Kristensen LE, et al. Prevalence of psoriatic arthritis in patients with psoriasis: A systematic review and meta-analysis of observational and clinical studies. *J Am Acad Dermatol* 2019;80(1):251-65.e19.
10. Cortese S, Moreira Maia CR, Rohde LA, et al. Prevalence of obesity in attention-deficit/hyperactivity disorder: study protocol for a systematic review and meta-analysis. *BMJ Open* 2014;4(3):e004541.
11. Holler JG, Bech CN, Henriksen DP, et al. Nontraumatic hypotension and shock in the emergency department and the prehospital setting, prevalence, etiology, and mortality: a systematic review. *PloS One* 2015;10(3):e0119331.
12. Hooi JKY, Lai WY, Ng WK, et al. Global Prevalence of Helicobacter pylori Infection: Systematic Review and Meta-Analysis. *Gastroenterology* 2017;153(2):420-9.